# Physicians’ Perceptions of Their Patients’ Attitude and Knowledge of Long-Term Oral Anticoagulant Therapy in Bulgaria

**DOI:** 10.3390/medicina55070313

**Published:** 2019-06-26

**Authors:** Nikolay Runev, Tatjana Potpara, Stefan Naydenov, Anita Vladimirova, Gergana Georgieva, Emil Manov

**Affiliations:** 1Department of Internal Diseases, Medical University of Sofia, 1431 Sofia, Bulgaria; 2Faculty of Medicine, University of Belgrade, 11000 Belgrade, Serbia; 3Boehringer Ingelheim RCV GmbH & Co KG Bulgarian Branch, 1505 Sofia, Bulgaria

**Keywords:** atrial, fibrillation, venous, thrombosis, anticoagulation, perception

## Abstract

*Background and Objectives*: Oral anticoagulation (OAC) is widely used in daily clinical practice worldwide for various indications. We aimed to explore the perception of Bulgarian clinicians about their patients’ attitude and knowledge of long-term OAC, prescribed for atrial fibrillation (AF) and/or known deep venous thrombosis (DVT)/pulmonary embolism (PE). *Materials and Methods*: We performed a cross-sectional study that involved 226 specialists: 187 (82.7%) cardiologists, 23 (10.2%) neurologists, and 16 (7.1%) vascular surgeons. They filled in a questionnaire, specially designed for our study, answering various questions regarding OAC treatment in their daily clinical practice. *Results*: The mean prescription rate of OACs in AF patients was 80.3% and in DVT/PE—88.6%. One hundred and eighty-seven (82.7%) of the participants stated they see their patients on OAC at least once per month. According to more than one-third of the inquired clinicians, the patients did not understand well enough the provided information concerning net clinical benefit of OAC treatment. About 68% of the clinicians declared that their patients would prefer a “mutual” approach, discussing with the physician the OAC options and taking together the final decision, whereas according to 43 (19.0%), the patients preferred the physician to take a decision for them. Patients’ OAC treatment had been interrupted at least once within the last year due to a physician’s decision by 178 (78.8%) of the participants and the most common reason was elective surgery. The most influential factors for a patient’s choice of OAC were the need of a specific diet to be kept, intake frequency, and possible adverse reactions. *Conclusions*: Our results suggest that a clinician’s continuous medical education, shared decision-making, and appropriate local strategies for improved awareness of AF/DVT/PE patients are key factors for improvement of OAC management.

## 1. Introduction

Oral anticoagulation (OAC) is widely used in contemporary clinical practice for preventive and therapeutic indications [1,2,3]. Until recently, the OAC treatment choice was restricted to the vitamin K antagonists (VKAs) and patients’ preferences were less commonly encountered [2,3,4]. Even after the introduction of new direct oral anticoagulants (DOACs) to clinical use and the large volume of data from clinical trials comparing VKAs’ and DOACs’ efficacy and safety, the patients’ awareness of the treatment choices, benefits, and risks with long-term OAC remains unclear [3,4,5,6,7]. Moreover, their role in the decision-making process is frequently neglected thus affecting the treatment adherence and persistence [5,7].

To date, there are no large-scale, population-based studies or national registries in Bulgaria evaluating the prescription rate of OACs and/or the prevalence of atrial fibrillation (AF), deep venous thrombosis (DVT)/pulmonary embolism (PE)—some of the most common diseases requiring OAC. A single center cross-sectional study, including 1027 patients, hospitalized for different diseases in the largest Bulgarian internal clinic showed that ~62% of all patients suffered at least one episode of AF [8]. Of these patients, ~14% had undergone ischemic stroke and OAC was prescribed to ~86% [8].

These data gave us grounds to perform the present study, aiming to explore the practicing clinicians’ perceptions of their patients’ attitude and knowledge of long-term OAC.

## 2. Materials and Methods

A cross-sectional questionnaire-based study involved 226 clinical specialists from 20 cities in Bulgaria. Of these, 187 (82.7%) were cardiologists, 23 (10.2%) neurologists, and 16 (7.1%) vascular surgeons. For the purposes of the study, a structured questionnaire with 12 questions was formulated, as shown in Appendix A.

The study was conducted from 1 July to 30 September 2017. The inclusion criteria were: (1) clinical experience for at least 5 years, and (2) regular prescription of OAC for atrial fibrillation (AF) and/or deep venous thrombosis (DVT)/pulmonary embolism (PE). The regular OAC prescription was defined as at least two prescriptions per week of a VKA or a DOAC. Exclusion criteria: (1) clinicians with other clinical specialties, (2) prescription of anticoagulants for indications other than those in the inclusion criteria, and (3) inability to fill in the study questionnaire for any reason.

As this is a non-trial activity (NTA) for health care providers (no involvement of patients at all in the study), an internal approval only has been done according to Boehringer Ingelheim requirements and standards (NTA tracking no. 170432). 

### Statistical Analysis

Categorical variables were shown as counts and percentages, and continuous variables as mean values and standard deviation (SD). Where normal distribution was not confirmed using the Shapiro–Wilk test, the median value and interquartile range (IQR) were used for variables with skewed distribution. Categorical variables were compared using the independent samples Chi-square test. Analysis of variance (ANOVA) was used for comparison of parametric data and the Mann–Whitney U test for non-parametric data. A two-sided *p*-value of <0.05 was considered statistically significant. The statistical analysis was performed by SPSS statistical package, version 19.0 (SPSS Inc., Chicago, IL, USA).

## 3. Results

The answers of the questions 1 to 7 are provided in Table 1 and Table 2. The vast majority of 226 participating physicians (210, 92.9%) reported that they treated patients with AF. The highest average monthly number of patients with AF was seen by cardiologists (46, 60.5%), whereas those with known DVT/PE by vascular surgeons (21, 60.0%). The physicians stated they had assigned an OAC to 80.3% of their AF patients for stroke prevention and to 88.6% of the patients with known DVT/PE for secondary prevention. One hundred and eighty-seven (82.7%) of the participants in our study declared they see their patients on OAC at least once per month, 28 (12.4%)—at least twice and 11 (4.9%)—three or more times per month (without significant difference between the specialties), as shown in Table 1.

According to 168 physicians (74.3%), the patients were satisfied with the provided general information about OAC treatment before its initiation, 42 (18.6%) answered their patients were very satisfied, and 16 (7.1%)—the patients were neither satisfied nor unsatisfied (no statistically significant difference between cardiologists, neurologists, and vascular surgeons). However, only 64.2% of the surveyed physicians answered that their patients understand well enough the discussed information about OAC, as shown in Table 1.

In terms of the patients’ role in the decision-making process concerning OAC, 153 (67.7%) of the participants stated that their patients preferred to discuss with the physician all possible treatment options (advantages and disadvantages) and then make a decision together (equal role of both sides), 43 (19.0%)—the patients preferred the physician to take a decision for them (with or without prior discussion, whereas 30 (13.3%) reported the patients would make their therapeutic choice alone irrespective of the physician’s opinion during the discussion, as shown in Table 2.

Patients’ OAC therapy had been interrupted at least once within the last 12 months due to a physician’s decision by 178 (78.8%). This occurred significantly more often with cardiologists and vascular surgeons and more rarely with neurologists. The most common reasons for interruption of OAC was the elective surgery—reported by 167 (93.8%), followed by an emergency—by 11 (6.2%) physicians, as shown in Table 2.

The physicians’ answers of the question “In your opinion, how do your patients rate the importance of treatment outcomes and attributes of OAC?” are shown in Figure 1.

The responders were asked to rank up to three factors (out of a total of six) which their patients would consider the most important in their choice of OAC. The drug’s efficacy was most commonly rated as the third important factor, whereas the need a specific diet to be kept, intake frequency, and possible adverse reactions were considered the most influential factors by the majority of the physicians, as shown in Figure 2.

The physicians were asked to rate five complications by importance and then to report how they evaluate their patients’ opinion about those complications. There was a close similarity between physicians’ answers and their opinions about what would the patients say when they chose an OAC. The complication the physicians rated the highest was stroke/DVT/PE, followed by bleeding, and then surgical emergencies, as shown in Figure 3a,b.

Most commonly, the information about the assigned OAC was provided to patients by the prescribing physician (223 physicians, 98.7%), followed by the printed leaflet inserted into the drug package (164, 72.6%), internet and online forums (125, 55.3%), friends/family (80, 35.4%), other patients (79, 35.0%), another physician (68, 30.1%), other materials supplied together with the drugs (47, 20.8%), and nurse/pharmacist (21, 9.3%), as shown in Figure 4.

## 4. Discussion

This is a cross-sectional questionnaire-based study, conducted among 226 physicians in 20 towns of Bulgaria, the vast majority of which 82.7% were cardiologists, 10.2% were neurologists, and 7.1% were vascular surgeons. They answered specific questions in order to evaluate the attitude, knowledge, and preferences of their patients with AF and/or DVT/PE concerning the long-term OAC treatment.

The prescription rate of oral anticoagulants in our study was relatively high compared the published data of other authors—80.3% of the physicians dealing with AF (mostly cardiologists) stated their patients had been treated with an OAC (VKA or DOAC) for prevention of embolic events [1,2,3,4]. Atrial fibrillation was the most common indication for prescription of OAC [7,8,9,10,11,12,13,14]. This could be explained by the high prevalence of this arrhythmia in the general population (25% long-life risk for at least one episode of AF in people aged >40 years) and probably by the significantly improved awareness of the cardiologists and other health specialists for the last two decades about the cardio-embolic risk of AF [11,12,15,16,17]. In developed countries, 65% to 80% of the patients diagnosed with AF and a CHADS_2_ or CHA_2_DS_2_VASc risk score higher than 2, receive long-term oral anticoagulation [2,3,15]. Data from the USA IMS Health’s National Disease and Therapeutic Index revealed that OAC prescription rates in AF patients with high thromboembolic risk varied from 20% to 80% [2]. Interestingly, ~15% of all AF patients in these studies did not receive any antithrombotic treatment because of various concomitant conditions (active bleeding, patient refusal, severe comorbid illnesses, pregnancy, etc.) [3,16,17,18,19]. In our study, the percentage of non-anticoagulated AF patients was around 20%. In contrast to other studies, we found similar (~67%) prescription rates of OAC (VKA or DOAC) for AF (most often by cardiologists) and DVT/PE (by vascular surgeons). Other authors reported a higher use of oral anticoagulants for DVT and/or PE, than in the case of AF—85–88% versus 60–80%, respectively [1,2,3,4,5]. It should be pointed out, the level of education and patients’ knowledge had a direct influence on the OAC management for embolic stroke prevention in AF [9,18,19,20,21].

Most of the physicians (~83%) of all specialties enrolled in our study stated they had visits with their patients on OAC therapy at least once per month. According to the guidelines, patients on DOACs should be seen at least once per year and if their kidney function was impaired (creatinine clearance <60 mL/min), the annual frequency of clinical visits could be calculated dividing the creatinine clearance by 10 [1,2].

The vast majority of our participants considered their patients were satisfied or very satisfied with the provided information about OACs. Since no patients were included in our study we could not discuss if the physician’s opinion about the patient’s level of satisfaction was really true. According to other studies, patients’ satisfaction varies widely from 42% to 97% [18,22,23].

The patient’s proper understanding of the provided information regarding the OAC treatment could influence the decision-making process, adherence to therapy, and the immediate and long-term prognosis, respectively [1,3]. In addition, there was frequently a discrepancy between the level of patient satisfaction from the medical information and the real understanding of this information [4,5]. According to only 64.2% of the inquired physicians in our study, their patients understood well enough the discussed information. In the study of Nadar et al., conducted in the United Kingdom, between 18% and 45% of all patients (depending on the ethnic group) had difficulties with the proper understanding about any aspect of the antithrombotic treatment [16].

Regarding the decision-making process, about 68% of our participants stated their patients would prefer a “mutual” approach, discussing with the physician the treatment options and taking together the final decision. Only 13% of the physicians answered the patients would prefer to decide themselves irrespective of the physician’s opinion. These data were supported by other studies, which confirmed that shared decision-making would improve patients’ adherence and persistence to OAC treatment [13,24,25].

An alarming finding of our study was that at least ~79% of all patients had stopped their OAC therapy at least once per year. These results were similar to other published data [2,5]. Our participants reported elective surgery as the most frequent cause for treatment cessation. According to other studies, gastrointestinal procedures and surgery with biopsy were the most frequent reason for temporary OAC discontinuation [1,3]. These findings were important from a clinical point of view—previous studies had shown that major thromboembolic complications, especially ischemic stroke, occurred in approximately 1% of the patients who underwent temporary discontinuation of OAC before an invasive procedure [2,3].

Intriguing answers were given by the participants concerning the most important characteristics of OAC medication according to their patients—the highest role of stroke/DVT/PE prevention, followed by the availability of a reversal agent in case of bleeding or emergencies. Obviously, the presence of a specific inhibitor of the OAC action make the patients more confident and trustworthy during a discussion of their anticoagulation management. This is particularly relevant for Dabigatran use in daily clinical practice [23].

In our study the most important factors for a patient’s choice about OAC therapy were the necessity to keep a specific diet, the intake frequency, and the possible adverse reactions. They were reported as “first three factors” exerting influence on the therapeutic decision by the majority of the participants. Surprisingly, the efficacy of the OAC drugs was classified most frequently as the third important factor by many of them. This finding could be related to patients’ misperception of OAC risks and benefits, which might be overcome by a well-structured patient education program [13]. However, in our study, the physicians’ answers and their opinions about the patients’ judgment of the complications, concerning the choice of OAC, were very similar—the highest importance was attributed to stroke/DVT/PE, followed by bleeding and surgical emergencies. Perhaps, the communication gap in this population had not been so large as expected based on the previously published data [20,25].

Nowadays, concerns about the patients’ awareness, choices, and preferences are becoming more important for the complex therapeutic approach, including antithrombotic treatment. Patients want to participate actively in the decision-making about the proposed procedures or treatments and to know all their alternatives [4,5]. The environment in which patients consume medical and health information has changed dramatically during the past decades worldwide [9,10]. The rapid diffusion of Internet technologies within the public sphere has placed an unprecedented amount of health information within reach of general consumers [6]. Nevertheless, according to our data, the prescribing physician continues to be a principal trusted resource for information about the OAC in spite of the variety of new sources of health information. Direct communication between physicians and patients is still an influential factor for the patient’s awareness and decisions. The second and third most used sources of information in our study—the printed leaflet (provided together with the box of the medication in Bulgaria) and internet/online forums—could exert positive but also negative influence (particularly the last source) on the patient’s attitude and therapeutic choice. We have the impression from our own clinical practice that the reading of the printed leaflet of the drugs (short product characteristics) might discourage the patients. Internet sites are not always reliable for the information they provide, and many could mislead the patients to an inappropriate decision. A study conducted in the USA among Hispanic adult patients with different diseases revealed that common health information sources were doctors (71%), television (68%), family and friends (63%), newspapers and magazines (51%), and radio (40%) [25]. In Europe, the Electrophysiology Wire Survey showed that considerable amount of time and resources were used to inform AF patients about their risk profile and appropriate management. However, a diversity of strategies across the European hospitals was reported [13].

*Study limitations*: (1) Disproportion of cardiologists, neurologists, and vascular surgeons in our study with the much larger number of the first ones. Patients with AF would attend a cardiologist for examinations and treatment (those without a cerebrovascular or peripheral arterial embolic event would not be referred to neurologist or a vascular surgeon). Patients with DVT were likely to visit a vascular surgeon for evaluation and treatment (a much fewer number in our study) and were less likely to be registered/followed up by a cardiologist or a neurologist unless another condition requiring OAC (AF, prosthetic valve disease, post-stroke, etc.) was present. (2) Our study did not include patients of the inquired physicians to answer the same questions, so we could not compare the answers of both sides. (3) The number of participants was not large enough for our results to be extrapolated nationwide.

## 5. Conclusions

This questionnaire-based study, conducted among Bulgarian cardiologists, neurologists, and vascular surgeons, shows that patients’ attitude and knowledge about OAC depends on their direct communication with the doctor rather than on drug leaflets or internet/online sources. Our patients prefer to discuss thoroughly with the physician the therapeutic options and then to take together the final decision. However, more than one-third of the physicians report their patients do not understand well enough the provided information concerning net clinical benefit of OAC. Moreover, the discontinuation of OAC remains an important issue in our population. These data suggest that physicians’ continuous medical education, shared decision-making, and appropriate local strategies for better informing AF/DVT/PE patients are the crucial factors for improvement of OAC management.

## Figures and Tables

**Figure 1 medicina-55-00313-f001:**
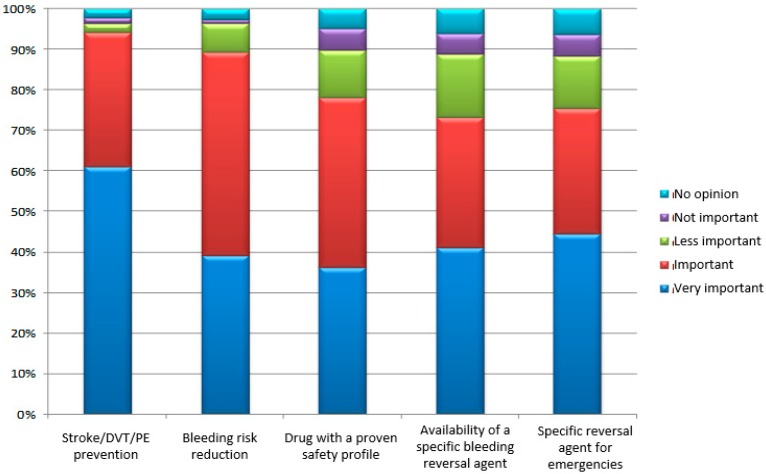
Some basic characteristics of the OAC treatment according to the patients (the information was provided by their physicians); DVT—deep venous thrombosis; PE—pulmonary embolism.

**Figure 2 medicina-55-00313-f002:**
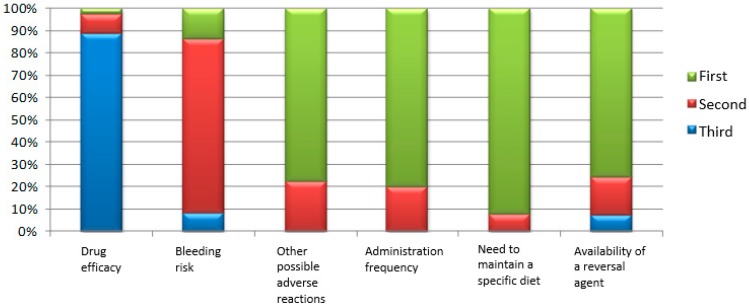
Rankings of factors of the surveyed physicians based on their responses to the question “If you had to discuss with your patients the OAC to be prescribed, which three factors they would consider the most important in your opinion?” (absolute number; relative share).

**Figure 3 medicina-55-00313-f003:**
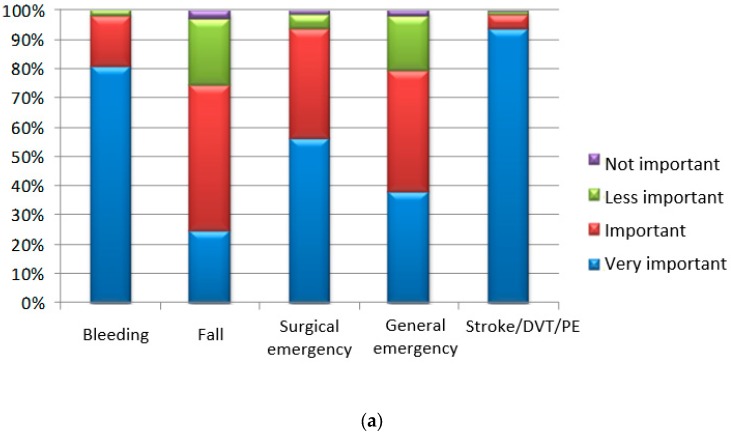
(**a**) Importance of five basic complications according to the physicians, when choosing an OAC. (**b**) Importance of five basic complications according to the patients, when choosing an OAC (the information was provided by their physicians); DVT—deep venous thrombosis; PE—pulmonary embolism.

**Figure 4 medicina-55-00313-f004:**
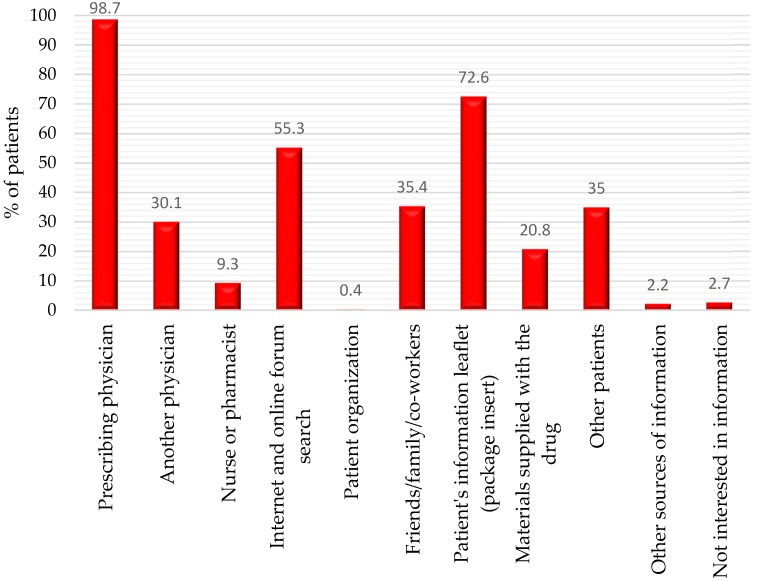
Sources of information from the surveyed physicians based on their responses to the question “What sources, in your opinion, do patients use to get information about the OAC they are taking?”.

**Table 1 medicina-55-00313-t001:** Questions 1 to 7 and the clinicians’ answers.

Question and Answers	All N (%)	Cardiologists N (%)	Neurologists N (%)	Vascular Surgeons N (%)	*p* Value
1. What is the average monthly number of patients with the following diagnoses in your practice?					
AF	76 (100%)	46 (60.5%)	16 (21.1%)	14 (18.4%)	*p* < 0.001
Known DVT/PE	35 (100%)	10 (28.6%)	4 (11.4%)	21 (60.0%)	*p* < 0.001
2. What is the approximate proportion of your patients with: AF treated with OAC for stroke prevention (%) or known deep vein thrombosis (DVT) with or without pulmonary embolism (PE) on OAC (%)?					
AF	61 (80.3%)	41 (67.2%)	11 (18.0%)	9 (14.8%)	*p* < 0.001
Known DVT/PE	31 (88.6%)	8 (25.8%)	2 (6.5%)	21 (67.7%)	*p* < 0.001
3. How often do you see your patients on OAC per month?					
At least once	187 (82.7%)	155 (82.9%)	19 (82.6%)	13 (81.3%)	NS
At least twice	28 (12.4%)	23 (12.3%)	3 (13.0%)	2 (12.5%)	NS
3 or more times	11 (4.9%)	9 (4.8%)	1 (4.3%)	1 (6.3%)	NS
4. In your opinion, how satisfied are your patients with the information about the OAC treatment they receive before its initiation?					
Satisfied	168 (74.3%)	139 (74.3%)	17 (73.9%)	12 (75.0%)	NS
Very satisfied	42 (18.6%)	35 (18.7%)	4 (17.4%)	3 (18.75%)	NS
Neither satisfied nor unsatisfied	16 (7.1%)	13 (7.0%)	2 (8.7%)	1 (6.25%)	NS
5. How do you rate your patients’ understanding of the provided information about OAC?					
Very good	101 (44.7%)	84 (44.9%)	10 (43.5%)	7 (43.8%)	NS
Good	44 (19.5%)	36 (19.3%)	5 (21.7%)	3 (18.8%)	NS
Inadequate	81 (35.8%)	67 (35.8%)	8 (34.8%)	6 (37.5%)	NS

AF—atrial fibrillation, DVT—deep venous thrombosis, PE—pulmonary embolism, OAC—oral anticoagulation, NS—non-significant; the *p* value refers to the inter-physicians’ answers.

**Table 2 medicina-55-00313-t002:** Questions 6 and 7, and the clinicians’ answers.

6. In your opinion, to what extent would your patients like to be involved in the choice of OAC?					
They prefer to discuss with the physician, then make the decision together.	153 (67.7%)	135 (72.2%)	8 (34.8%)	10 (62.5%)	*p* = 0.02
They prefer to discuss with the physician, then make the decision by themselves.	30 (13.3%)	25 (13.4%)	2 (8.7%)	3 (18.75%)	*p* < 0.001
They prefer the physician to make the decision.	43 (19.0%)	27 (14.4%)	13 (56.5%)	3 (18.75%)	*p* < 0.001
7. Did you have to stop the current OAC in any of your patients within the past 12 months due to a planned surgery and/or emergency?					
Yes, I had to stop OAC	178 (78.8%)	151 (80.7%)	14 (60.9%)	13 (81.3%)	*p* < 0.001
Due to a planned surgery	167 (93.8%)	146 (96.7%)	8 (57.1%)	13 (100%)	*p* = 0.02
Because of an emergency	11 (6.2%)	5 (3.3%)	6 (42.9%)	0	*p* < 0.01

OAC—oral anticoagulation; the *p* value refers to the inter-physicians’ answers.

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
