# Peer review of "Physicians’ Perceptions of Their Patients’ Attitude and Knowledge of Long-Term Oral Anticoagulant Therapy in Bulgaria"

_medicina, 2019, doi:10.3390/medicina55070313_

Round 1

Reviewer 1 Report

In the manuscript titled "Physicians perceptions of their patients’ attitude and knowledge on long-term oral anticoagulant therapy in Bulgaria", the authors have conducted a cross-sectional study to imprint the physician perception and their patient feedback on OAC treatment in their daily clinical practice. A total of 226 specialists were included. Overall, the manuscript is well designed and written. The information presented is clear and the derived results are very informative for the OAC treatment.

I only have some minor comments that can improve the quality of the manuscript.

1. Please add in introduction any relevant information on the percentage of Bulgarian population using oral anticoagulants. If no data is available, it would be interesting to mention the proportion of population diagnosed with atrial fibrillation or VTE, so that the reader clearly understands that OACs are commonly used drugs.

2. It is not clear whether both coumarinic anticoagulants and direct oral anticoagulants were prescribed by the interviewed physicians. The reader gets the impression that this study mostly concers coumarinic anticoagulants. Please discuss this point.

3. Please clarify for which comparisons the p value corresponds. Is it for the inter-physicians answers?

4. Table 2: It seems that a heading is missing

5. Line 229: Reference 20 is as superscript, please check for consistency in your reference list.

Author Response

Dear Reviewer,

Thank you very much for your comments and recommendations regarding our manuscript. They would definitely improve the quality of our work.

Below, we apply the answers to your questions/suggestions:

Reviewer 1: Please add in introduction any relevant information on the percentage of Bulgarian population using oral anticoagulants. If no data is available, it would be interesting to mention the proportion of population diagnosed with atrial fibrillation or VTE, so that the reader clearly understands that OACs are commonly used drugs.

Answer: There are no large-scale, population-based studies in Bulgaria evaluating the prescription rate of OACs and/or the prevalence of atrial fibrillation (AF), deep venous thrombosis (DVT)/pulmonary embolism (PE). We have added information from a relatively large cross-sectional, hospital-based Bulgarian study, evaluating the prevalence of AF and prescription of OAC for this indication. It was published in Medicina-Lithuania in 2018.

Reviewer 1: It is not clear whether both coumarinic anticoagulants and direct oral anticoagulants were prescribed by the interviewed physicians. The reader gets the impression that this study mostly concers coumarinic anticoagulants. Please discuss this point.

Answer: By OAC we mean both VKA and DOAC. We added this information in the inclusion criteria in line 62, subheading of tables 1 and 2, and in the discussion (lines 155 and 166-167)

Reviewer 1: Please clarify for which comparisons the p value corresponds. Is it for the inter-physicians answers?

Answer: The p values refers to the inter-physicians’ answers. We added this clarifying information in the text accordingly (under the tables).

Reviewer 1: Table 2: It seems that a heading is missing

Answer: The heading of Table 2 is present above the table. It is in line 108 - “Questions 6 and 7, and the clinicians’ answers”

Reviewer 1: Line 229: Reference 20 is as superscript, please check for consistency in your reference list.

Answer: We have corrected it

Additionally, we re-arranged the citation number the references accordingly after adding one more reference (N 8)

Yours,

Dr. Stefan Naydenov

corresponding author

Reviewer 2 Report

In this article, the author tries to figure out the practicing clinicians’ perceptions of their patients’ attitude and knowledge on long-term OAC. Since the awareness of the treatment choices for patients remains unclear and whether this issue will affect the treatment adherence and persistence remains unclear, this study provides a scientific based result for the future clinical treatment. Besides, this questionnaire-based study had involved a great number of clinical specialists from different cities in Bulgaria which enhanced the credibility of this study. Moreover, the writing is both clear and easy to understand, the tables and the figures are friendly to read. From their results, they had found that physicians’ continuous medical education and shared decision-making for better informing the patients are the major factors for improvement of OAC management. These new findings are valuable for future clinical usage. In the discussion, the author has thoroughly compared their study with other research such as why their prescription rate of OAC was relatively high compared with other studies. Besides, the patients’ and physicians categories, the proportion of different factors like patients’ satisfaction or decision-making preference, and the percentage of temporary OAC discontinuation are all well discussed in their study. Last by not least, the author has declared the study limitations clearly in the content.

Author Response

Dear Reviewer,

Thank you very much for your time and assistance evaluating our manuscript. Your review, highlighting the most important data of our article, was of great support to us.

Yours,

Dr. Stefan Naydenov
